# Body mass index and all-cause mortality in a 21st century U.S. population: A National Health Interview Survey analysis

**Aayush Visaria** [1] *, **Soko Setoguchi** [1,2]

**1** Rutgers Institute for Health, Center for Pharmacoepidemiology and Treatment Sciences, New Brunswick, NJ, United States of America, **2** Department of Medicine, Rutgers Robert Wood Johnson Medical School, New Brunswick, NJ, United States of America

* aayush.visaria@rutgers.edu

**Data Availability Statement:** The datasets generated and analyzed during the current study are publicly available at the National Health Interview Survey (NHIS) official website: https://

## Abstract

### Introduction

Much of the data on BMI-mortality associations stem from 20th century U.S. cohorts. The purpose of this study was to determine the association between BMI and mortality in a contemporary, nationally representative, 21st century, U.S. adult population.

### Methods

This was a retrospective cohort study of U.S. adults from the 1999–2018 National Health Interview Study (NHIS), linked to the National Death Index (NDI) through December 31st, 2019. BMI was calculated using self-reported height & weight and categorized into 9 groups. We estimated risk of all-cause mortality using multivariable Cox proportional hazards regression, adjusting for covariates, accounting for the survey design, and performing sub-group analyses to reduce analytic bias.

### Results

The study sample included 554,332 adults (mean age 46 years [SD 15], 50% female, 69% non-Hispanic White). Over a median follow-up of 9 years (IQR 5–14) and maximum follow-up of 20 years, there were 75,807 deaths. The risk of all-cause mortality was similar across a wide range of BMI categories: compared to BMI of 22.5–24.9 kg/m², the adjusted HR was 0.95 [95% CI 0.92, 0.98] for BMI of 25.0–27.4 and 0.93 [0.90, 0.96] for BMI of 27.5–29.9. These results persisted after restriction to healthy never-smokers and exclusion of subjects who died within the first two years of follow-up. A 21–108% increased mortality risk was seen for BMI ≥30. Older adults showed no significant increase in mortality between BMI of 22.5 and 34.9, while in younger adults this lack of increase was limited to the BMI range of 22.5 to 27.4.

### Conclusion

The risk of all-cause mortality was elevated by 21–108% among participants with BMI ≥30. BMI may not necessarily increase mortality independently of other risk factors in adults,

www.cdc.gov/nchs/nhis/nhis_2011_data_release.
htm.

**Funding:** The author(s) received no specific
funding for this work.

**Competing interests:** The authors have declared
that no competing interests exist.

especially older adults, with overweight BMI. Further studies incorporating weight history,
body composition, and morbidity outcomes are needed to fully characterize BMI-mortality
associations.

## Introduction

The U.S. prevalence of overweight and obesity has risen dramatically over the last 25 years,
with more than half of all adults in the U.S. now overweight or obese [1]. It is well-established
that elevated BMI can contribute to several cardio-metabolic conditions, including diabetes,
hypertension, and coronary artery disease, which are among the leading causes of premature
death in the U.S. [2].

A number of studies have examined the association between BMI and mortality in the gen-
eral U.S. population, including the landmark Cancer Prevention Study I and II cohorts [3] and
other National Cancer Institute Cohort Consortium studies [4]. Furthermore, in the most
comprehensive pooled analysis of U.S. data to date, the Global BMI Mortality Collaboration
conducted a participant-level meta-analysis of data from 45 cohort studies from the U.S. [5].
Nevertheless, epidemiologic evidence regarding the association between BMI and all-cause
mortality has been inconsistent, especially with regards to overweight and class I obese individ-
uals, with some meta-analyses demonstrating similar or lower risk of all-cause mortality [6,7]
and others finding significantly elevated mortality risk in individuals with BMI >25 [5,8,9]. In
addition, most U.S. studies to date have used data from the 1960s through the 1990s and have
included predominantly non-Hispanic White men and women. In contrast, the contemporary
U.S. population: (1) has a substantially different BMI distribution, with mean BMI having
risen by more than 2 kg/m$^2$ in both men and women since the 1970s [1,10], and increasing
skewness towards obese-range BMIs [10]; (2) has seen >10 year increases in life expectancy
both overall and among obese individuals [11,12]; (3) is more racially and ethnically diverse,
with the percentage of non-Hispanic Whites decreasing from 84% in 1990 to 58% in 2020 [13];
and (4) has seen improvements in efficacy and access to treatment strategies for obesity-related
conditions [14]. All of these factors may alter the association between BMI and mortality. Of
the few studies using more contemporary populations [15–18], they were either restricted by
small sample sizes [15–17] limiting analysis in racial and gender subgroups, and/or only rudi-
mentarily characterized the dose-response relationship between BMI and mortality [17]. Fur-
thermore, studies inconsistently adjusted for methodologic bias including confounding by
illness-related weight loss, collider bias (e.g. the concept that conditioning on obesity-related
disease may distort associations between risk factors like BMI and diet and subsequently bias
downstream associations), and the healthy person effect (e.g. selection bias that may be intro-
duced when selecting participants of different BMI groups who may otherwise be healthy)
[15,16].

To address this gap, we examined the association between BMI and all-cause mortality in
the general U.S. population using nationally representative data from the 1999–2018 National
Health Interview Survey (NHIS).

## Methods

### Data sources and study population

The NHIS is a nationally representative survey of the civilian, noninstitutionalized U.S. popu-
lation, following a multi-stage, probability design from 1999–2018. Further information about

the dataset and the sample design can be found in the supplement and elsewhere [19,20]. Because this dataset is publicly available, de-identified, and population-based, this study was considered non-Human Subjects research and exempt from Rutgers Institutional Review Board approval. During the study period, the National Center for Health Statistics (NCHS) maintained IRB approval from the NCHS Research Ethics Review Board (ERB).

We included non-pregnant adults ≥20 years old with recorded BMI. We excluded participants with missing data on covariates (see **S1 Appendix**, **Supplementary Methods**) and outlier BMI values (<10, ≥99).

## BMI, mortality, and covariates

BMI was calculated using self-reported weight and height (weight in kg/[height in m]$^2$). To allow for comparability with previous studies and examination of non-linear associations, BMI was classified into 9 categories, consistent with the large, pooled analysis of National Cancer Institute Cohort Consortium studies: <18.5, 18.5–19.9, 20.0–22.4, 22.5–24.9, 25–27.4, 27.5–29.9, 30.0–34.9, 35.0–39.9, ≥40 kg/m$^2$ [4].

All-cause mortality was determined from the U.S. National Death Index (NDI). The 1999–2018 NHIS was linked by the National Center for Health Statistics to the NDI up to December 31$^{st}$, 2019 via patient unique identifiers [21].

We selected covariates a priori based on clinical knowledge and previous studies of BMI and mortality. These included demographics (age, gender, race/ethnicity), socio-behavioral factors (education, marital status, physical activity, smoking status, alcohol consumption, insurance coverage, region of residence, citizenship status), comorbidities (self-reported history of cardiovascular disease, non-skin cancer or melanoma, COPD, current asthma, liver disease, kidney disease, diabetes, or functional limitations), and healthcare utilization factors (doctor's visit in the past 12 months, mental health visit in the past 12 months). Covariate categorizations are provided in the supplement (Extended Supplementary Methods). Questionnaire protocols and quality control procedures for each covariate are described elsewhere [20].

## Statistical analysis

NHIS baseline characteristics were compared across the 9 specified BMI categories. To determine the association between BMI and all-cause mortality, we used Cox proportional hazard models (proportionality assumption verified using Schoenfeld residual plots), adjusting for covariates selected a priori and using BMI of 22.5–24.9 as the reference category. As only the year and quarter were available for death date and interview date, we used the difference between the year and quarter of the death date or end-of-follow-up and survey interview year and quarter to calculate time-to-event. Participants who died or reached the end of follow-up were censored. Cox models were adjusted for age, gender, race/ethnicity, education, marital status, physical activity, alcohol consumption, insurance coverage, region of residence, and citizenship status. Consistent with prior studies, we did not adjust for comorbidities in the main analysis, as they could be on the causal pathway between BMI and mortality or potential colliders for BMI and unmeasured confounders [3–5] (**S1 Fig in S2 Appendix**). However, we performed a sensitivity analysis adjusting for metabolic syndrome criteria and additionally excluding participants with any reported chronic disease to explore the independent relationship between BMI and all-cause mortality.

To understand the association between BMI and mortality in specific populations, we conducted analyses in women aged <65, women ≥65, men <65, and men ≥65, as prior studies have suggested weaker BMI-mortality associations in women and older adults. We also performed analyses stratified by race/ethnicity (non-Hispanic White, non-Hispanic Black, non-

Hispanic Asian, Hispanic), as different ethnic groups exhibit differences in BMI distributions and associations with morbidity. We further conducted analyses among subgroups of disease-free individuals, utilizing different definitions of disease: minor morbidity, defined as self-reported history of cardiovascular disease, non-skin cancer except melanoma, COPD, current asthma, liver disease, kidney disease, diabetes, or functional limitations; and major morbidity, defined via Berrington de Gonzalez et al. [4] as self-reported history of cardiovascular disease and non-skin cancer or melanoma.

We also conducted analyses restricted to healthy never-smokers (no self-reported cardio-vascular disease or non-skin cancer or melanoma) to address confounding by major chronic disease and smoking [4]. To examine the potential impact of reverse causality (i.e. confounding by illness-related weight loss), we conducted analyses limited to participants who did not die within the first two years of follow-up. Although some prior meta-analyses [5] have excluded the first five years of follow-up, there has been concern that such exclusions would lead to significant selection bias as a sizeable proportion of the study population would be excluded, without significant benefit in reducing reverse causality [22]. Nevertheless, we conducted sensitivity analyses excluding the first five years of follow-up.

### Sensitivity analyses using NHANES

Limitations of the NHIS include use of self-reported data and lack of laboratory data and information on waist circumference, change in weight over time, and other potential confounding variables including additional comorbidities and dietary factors. To address these limitations, we estimated associations between BMI and all-cause mortality in the 1999–2018 National Health and Nutrition Examination Surveys (NHANES) stratified by waist circumference, weight change, and using alternate definitions to define healthy individuals. Further information about the NHANES study population can be found elsewhere [23] and in the supplement.

Both NHIS and NHANES analyses utilized weights to produce nationally representative estimates and to account for oversampling of certain groups including older adults and members of ethnic minorities. All analyses also accounted for the complex survey design with strata and cluster variables using a traditional Taylor linearization approach to obtain accurate variance estimates. Analyses were conducted using SAS version 9.4 (SAS Institute Inc., Cary, NC) with a two-sided significance level of 0.05.

## Results

### Sample characteristics

The study sample included 554,332 adults (mean age 46 years, 50% female, 69% non-Hispanic White, 12% non-Hispanic Black, 42% smoking at least 100 cigarettes in their life). The mean BMI was 27.5 (SD 6.1). Between the first four-year cycle (1999–2002) and the last cycle (2015–2018), mean BMI rose from 26.7 to 28.0 kg/m$^2$ and prevalence of BMI of $\geq$30 increased from 22% to 31% (p<0.001 for trend; see **S2 Fig in S2 Appendix** for distribution of BMI). Nearly 21% had BMI of 25.0–27.4, and 14% had BMI of 27.5–29.9 (**Table 1**). Compared to participants with BMI of 22.5–24.9, those with BMI $\geq$40 were more likely to be female (63% vs. 52%), non-Hispanic Black (21% vs. 9.6%), and U.S. citizens (96% vs. 91%) (**Table 1**). Individuals with a BMI of 30–34.9 had more than double the proportion of diabetes (13% vs. 4.3%) and nearly double the proportion of MI (4.2% vs. 2.6%) and hypertension (39% vs. 20%) compared to a BMI of 22.5–24.9. Participants with BMI <18.5 also had a high comorbidity burden, including 7.8% (vs. 6.0%) with non-skin malignancy, 9.8% (vs. 4.6%) with COPD, and 11% (vs. 3.6%) with functional limitations.

**Table 1. Baseline characteristics by BMI category among NHIS 1999–2018 participants.**

| NHIS 1999–2018 | | Body Mass Index (BMI) Category | | | | | | | | |
|---|---|---|---|---|---|---|---|---|---|---|
| Variables | Overall | <18.5 kg/m2 | 18.5–19.9 | 20.0–22.4 | 22.5–24.9 | 25.0–27.4 | 27.5–29.9 | 30.0–34.9 | 35.0–39.9 | ≥40 |
| **Number of Subjects** | 554,332 | 10,316 (1.9%) | 21,648 (4.0%) | 74,485 (14%) | 103,441 (19%) | 115,601 (21%) | 77,835 (14%) | 93,002 (17%) | 35,347 (6.3%) | 22,657 (3.9%) |
| **Demographics/Sociobehavioral Factors** | | | | | | | | | | |
| **Age, median (IQR)** | 44 (31–58) | 36 (23–59) | 35 (23–52) | 39 (25–54) | 43 (29–58) | 46 (33–60) | 47 (34–59) | 47 (35–59) | 46 (34–57) | 45 (33–56) |
| > = 65 (%) | 120,742 (18%) | 2890 (21%) | 4364 (15%) | 15066 (16%) | 23561 (19%) | 26972 (19%) | 17869 (19%) | 20206 (18%) | 6533 (15%) | 3281 (11%) |
| **Sex (% Female)** | 301790 (50%) | 8020 (74%) | 16761 (75%) | 49461 (64%) | 56949 (52%) | 52421 (41%) | 34284 (39%) | 47796 (46%) | 20684 (52%) | 15414 (63%) |
| **Race/Ethnicity** | | | | | | | | | | |
| **Non-Hispanic White** | 354870 (69%) | 6905 (70%) | 14811 (73%) | 50559 (72%) | 68210 (71%) | 74363 (70%) | 48788 (68%) | 57127 (67%) | 20990 (65%) | 13117 (63%) |
| **Non-Hispanic Black** | 77647 (12%) | 1063 (9.1%) | 2095 (8.5%) | 7666 (8.8%) | 11720 (9.6%) | 14678 (10%) | 11558 (12%) | 15877 (14%) | 7380 (17%) | 5610 (21%) |
| **Hispanic** | 90071 (14%) | 1028 (8.5%) | 2283 (8.5%) | 9539 (10%) | 16141 (13%) | 20379 (15%) | 14396 (16%) | 17021 (16%) | 5970 (15%) | 3314 (13%) |
| **Non-Hispanic Asian** | 25371 (4.6%) | 1210 (11%) | 2230 (9.5%) | 6038 (7.8%) | 6406 (6.3%) | 5053 (4.5%) | 2181 (2.9%) | 1685 (1.9%) | 389 (1.1%) | 179 (0.7%) |
| **Native American/Multiracial/Other** | 6373 (1.1%) | 110 (0.9%) | 229 (0.9%) | 683 (0.8%) | 964 (0.8%) | 1128 (0.9%) | 912 (1.0%) | 1292 (1.3%) | 618 (1.7%) | 437 (1.9%) |
| **Education** | | | | | | | | | | |
| **High School** | 316144 (54%) | 6263 (58%) | 11938 (53%) | 40984 (52%) | 57736 (53%) | 66134 (54%) | 45010 (54%) | 54628 (56%) | 20419 (56%) | 13032 (56%) |
| **Bachelor's degree** | 197147 (38%) | 3355 (34%) | 7880 (38%) | 26947 (39%) | 36621 (38%) | 40250 (37%) | 27439 (38%) | 32993 (38%) | 13064 (38%) | 8598 (40%) |
| **Graduate Degree or Higher** | 41041 (7.9%) | 698 (6.6%) | 1830 (8.6%) | 6554 (9.2%) | 9084 (9.5%) | 9217 (8.7%) | 5386 (7.6%) | 5381 (6.2%) | 1864 (5.5%) | 1027 (4.6%) |
| **Citizenship Status** | | | | | | | | | | |
| **U.S. Citizen (%)** | 506092 (92%) | 9195 (90%) | 19546 (91%) | 21648 (92%) | 93494 (91%) | 104285 (91%) | 70796 (92%) | 86005 (93%) | 33516 (95%) | 21776 (96%) |
| Region of Residence (N = 27965 missing from survey cycle 2004) | | | | | | | | | | |
| **Northeast** | 88716 (18%) | 1601 (18%) | 3515 (18%) | 12508 (19%) | 17418 (19%) | 19131 (18%) | 12355 (18%) | 14081 (17%) | 5024 (16%) | 3083 (15%) |
| **Midwest** | 118320 (24%) | 2054 (23%) | 4457 (24%) | 15243 (23%) | 21261 (23%) | 24174 (24%) | 16727 (24%) | 10841 (25%) | 8072 (26%) | 5491 (27%) |
| **South** | 192678 (37%) | 3653 (38%) | 7264 (35%) | 24511 (35%) | 34537 (35%) | 39331 (36%) | 27224 (37%) | 33675 (38%) | 13596 (40%) | 8887 (41%) |
| **West** | 126653 (22%) | 2477 (22%) | 5276 (23%) | 18271 (23%) | 24712 (22%) | 27038 (22%) | 17614 (21%) | 19936 (20%) | 7103 (19%) | 4226 (17%) |
| **Smoking Status (Smoked>100 cigarettes?)** | 235349 (42%) | 4592 (42%) | 8590 (38%) | 29950 (39%) | 42874 (40%) | 49625 (42%) | 34093 (43%) | 40818 (44%) | 15348 (43%) | 9459 (41%) |
| **Marital Status** | | | | | | | | | | |
| **Never Married** | 124060 (21%) | 3213 (36%) | 6624 (32%) | 21442 (28%) | 24482 (22%) | 22986 (18%) | 14155 (16%) | 17353 (16%) | 7624 (19%) | 6181 (24%) |
| **Separated/Divorced/Widowed** | 156230 (18%) | 3312 (21%) | 5929 (18%) | 19545 (17%) | 28700 (18%) | 32202 (18%) | 21936 (18%) | 26926 (19%) | 10704 (20%) | 6976 (22%) |
| **Married** | 242740 (54%) | 3222 (36%) | 7726 (43%) | 29034 (47%) | 44347 (53%) | 54104 (58%) | 37452 (60%) | 43491 (59%) | 15055 (55%) | 8309 (48%) |
| **Other** | 31302 (6.8%) | 569 (6.7%) | 1369 (7.9%) | 4464 (7.3%) | 5912 (6.7%) | 6309 (6.4%) | 4292 (6.3%) | 5232 (6.4%) | 1964 (6.4%) | 1191 (5.8%) |

*(Continued)*

**Table 1.** (Continued)

| NHIS 1999–2018 | | Body Mass Index (BMI) Category | | | | | | | | |
|---|---|---|---|---|---|---|---|---|---|---|
| **No alcohol consumption (%)** | 87271 (16%) | 2307 (25%) | 3960 (20%) | 12312 (18%) | 15799 (16%) | 16803 (14%) | 11347 (14%) | 14648 (16%) | 5901 (17%) | 4194 (19%) |
| **Physical Activity (Moderate-level activity minutes)** | | | | | | | | | | |
| **0 MEMs (Inactive)** | 210336 (36%) | 5051 (46%) | 8033 (37%) | 25449 (32%) | 35704 (32%) | 41232 (35%) | 29324 (37%) | 38404 (41%) | 15794 (44%) | 11345 (48%) |
| **0–150 MEMs (Insufficiently Active)** | 97534 (18%) | 1710 (17%) | 3634 (17%) | 12122 (17%) | 16959 (16%) | 19472 (17%) | 13901 (18%) | 17686 (19%) | 7239 (20%) | 4811 (22%) |
| **150–300 MEMs (Sufficiently Active)** | 81385 (15%) | 1238 (12%) | 3218 (15%) | 11521 (16%) | 15696 (16%) | 17659 (16%) | 11443 (15%) | 13225 (14%) | 4745 (14%) | 2640 (12%) |
| **>300 MEMs (Highly Active)** | 165077 (31%) | 2317 (24%) | 6763 (32%) | 25393 (35%) | 35082 (36%) | 37238 (32%) | 23167 (30%) | 23687 (26%) | 7569 (22%) | 3861 (18%) |
| **Strength Training** | | | | | | | | | | |
| **Yes (≥2 times/week)** | 91434 (17%) | 1258 (13%) | 3753 (18%) | 14471 (20%) | 20408 (21%) | 20949 (19%) | 12657 (17%) | 12125 (14%) | 3872 (12%) | 1941 (8.9%) |
| Health Service Factors | | | | | | | | | | |
| **Insurance Coverage (%)** | 469533 (85%) | 8691 (84%) | 18120 (84%) | 62806 (85%) | 87798 (86%) | 98077 (86%) | 65955 (86%) | 78851 (85%) | 30022 (85%) | 19213 (85%) |
| **Doctor's Visit in Past Year (%)** | 426906 (77%) | 7806 (75%) | 16440 (76%) | 55879 (75%) | 77786 (76%) | 87120 (76%) | 59756 (77%) | 73758 (80%) | 29185 (83%) | 19176 (85%) |
| **Mental Health Visit in Past 12 Months** | 43076 (7.4%) | 934 (8.5%) | 1911 (8.5%) | 5832 (7.7%) | 7157 (6.6%) | 7783 (6.4%) | 5548 (6.8%) | 7613 (7.8%) | 3503 (9.4%) | 2795 (12%) |
| Comorbidities | | | | | | | | | | |
| **Cardiovascular Disease (%)** | 61849 (13%) | 1362 (14%) | 2067 (10%) | 6752 (10%) | 10421 (11%) | 12400 (13%) | 9112 (14%) | 11539 (15%) | 4785 (17%) | 3411 (19%) |
| **History of Stroke** | 13113 (2.6%) | 389 (3.9%) | 497 (2.2%) | 1450 (2.0%) | 2196 (2.3%) | 2583 (2.4%) | 1946 (2.8%) | 2396 (2.9%) | 978 (3.3%) | 678 (3.5%) |
| **History of Myocardial Infarction** | 16224 (3.4%) | 326 (3.4%) | 404 (1.9%) | 1461 (2.1%) | 2490 (2.6%) | 3429 (3.4%) | 2600 (3.9%) | 3253 (4.2%) | 1378 (5.0%) | 883 (4.9%) |
| **History of Coronary Heart Disease** | 20774 (4.3%) | 358 (3.7%) | 493 (2.4%) | 1808 (2.5%) | 3265 (3.6%) | 4388 (4.5%) | 3320 (5.1%) | 4240 (5.5%) | 1738 (6.1%) | 1164 (6.6%) |
| **History of Other Heart Disease** | 34828 (7.5%) | 788 (8.5%) | 1303 (6.7%) | 4135 (6.5%) | 5952 (6.8%) | 6769 (7.0%) | 4921 (7.6%) | 6322 (8.2%) | 2636 (9.4%) | 2002 (12%) |
| **Diabetes (%)** | 38621 (8.0%) | 292 (2.9%) | 455 (2.1%) | 1992 (2.9%) | 4134 (4.3%) | 6345 (6.1%) | 5924 (8.8%) | 9898 (13%) | 5279 (18%) | 4302 (25%) |
| **Hypertension (%)** | 131439 (28%) | 1529 (15%) | 2611 (12%) | 10112 (14%) | 18473 (20%) | 25709 (26%) | 20993 (32%) | 29708 (39%) | 13033 (47%) | 9271 (53%) |
| **Kidney Disease (%)** | 8662 (1.7%) | 261 (2.8%) | 340 (1.7%) | 902 (1.3%) | 1326 (1.3%) | 1562 (1.5%) | 1185 (1.6%) | 1627 (2.0%) | 802 (2.6%) | 657 (3.5%) |
| **Asthma (%)** | 28996 (6.5%) | 519 (6.3%) | 963 (5.5%) | 3176 (5.3%) | 4231 (5.1%) | 4921 (5.3%) | 3846 (6.2%) | 5732 (7.7%) | 2901 (10%) | 2707 (15%) |
| **Chronic Obstructive Pulmonary Disease (COPD) (%)** | 35729 (5.8%) | 1129 (9.8%) | 1343 (5.5%) | 3921 (4.7%) | 5250 (4.6%) | 6068 (4.6%) | 4689 (5.4%) | 6933 (6.8%) | 3452 (9.0%) | 2944 (12%) |
| **Liver condition (%)** | 6543 (1.3%) | 136 (1.5%) | 210 (1.0%) | 713 (1.0%) | 941 (1.0%) | 1266 (1.2%) | 905 (1.3%) | 1348 (1.7%) | 584 (2.1%) | 440 (2.2%) |
| **Non-Skin Cancer/Malignancy (%)** | 39065 (6.4%) | 757 (7.8%) | 1084 (5.2%) | 3689 (5.5%) | 5353 (6.0%) | 6078 (6.2%) | 4183 (6.3%) | 5080 (6.6%) | 1862 (6.6%) | 1194 (6.5%) |
| **Functional Limitations (%)** | 27665 (5.1%) | 1049 (11%) | 1007 (4.5%) | 2770 (3.7%) | 3738 (3.6%) | 4234 (3.7%) | 3266 (4.3%) | 5416 (6.2%) | 2983 (9.5%) | 3202 (17%) |
| **Depressive Symptoms (%)** | 15589 (3.1%) | 491 (5.3%) | 678 (3.3%) | 1755 (2.6%) | 2374 (2.5%) | 2553 (2.5%) | 1909 (2.7%) | 2998 (3.6%) | 1522 (5.1%) | 1309 (7.1%) |
| **Disease Status per Cancer Prevention Cohort Study II (%)** | 114155 (25%) | 2528 (28%) | 4001 (21%) | 13288 (20%) | 19235 (22%) | 22214 (23%) | 16208 (25%) | 21092 (28%) | 8972 (32%) | 6617 (37%) |

*(Continued)*

**Table 1.** (Continued)

| NHIS 1999–2018 | | Body Mass Index (BMI) Category | | | | | | | | |
|---|---|---|---|---|---|---|---|---|---|---|
| NHIS Disease* (%) | 197353 (42%) | 3281 (35%) | 5526 (28%) | 19558 (30%) | 31026 (35%) | 39003 (40%) | 29755 (46%) | 40248 (54%) | 17009 (61%) | 11947 (69%) |

*NHIS Disease definition: Presence of any one of the following: Self-reported asthma, COPD, emphysema, chronic bronchitis, non-skin cancer, current liver disease, cardiovascular disease (CAD, HF, stroke, MI), diabetes, hypertension, kidney disease, or functional limitation.

**Disease definition of the Cancer Prevention Cohort Study II includes presence of current asthma, cardiovascular disease, COPD, and non-skin cancer. Unintentional weight loss of at least 10 pounds, which was also included in the original disease definition, was not included in this study as there is no NHIS equivalent.

## Association between BMI and mortality

We observed 75,807 deaths during a median follow-up of 9 (IQR, 5–14 years; range, 0–20) years. Crude 5-year mortality rates per 1,000 person-years ranged from 10.7 among those with BMI of 30–34.9 to 35.2 per 1,000 person-years in those with BMI <18.5 (**S1 and S2 Tables in S2 Appendix**). The unadjusted risks of all-cause mortality were similar across BMI from 20.0 to 29.9 kg/m$^2$ (unadjusted HR [95% CI]; BMI 20.0–22.4: 0.96 [0.93, 0.99], BMI 22.5–24.9: 1.00 [Ref], BMI 25.0–27.4: 1.03 [1.00, 1.06], BMI 27.5–29.9: 1.01 [0.98, 1.04]) but were significantly elevated in participants with BMI of 30–34.9 (1.08 [1.04, 1.11]), BMI of 35–39.9 (1.12 [1.07, 1.16]), BMI ≥40 (1.31 [1.24, 1.37]), and BMI <18.5 (1.90 [1.79, 2.01]). These patterns became more apparent after adjustment for covariates (**Fig 1A**) and after further restricting the cohort to healthy (without non-skin cancer or melanoma or cardiovascular disease at baseline), never-smokers who did not die within the first two years of follow-up (**Fig 1B**). Upon additionally controlling for comorbidities including diabetes and hypertension (potential colliders), risk of mortality marginally decreased among overweight BMI categories (BMI 20.0–22.4: 1.10 [1.03,1.19], BMI 22.5–24.9: 1.00 [Ref], BMI 25.0–27.4: 0.95 [0.89,1.00], BMI 27.5–29.9: 0.96 [0.90, 1.02]).

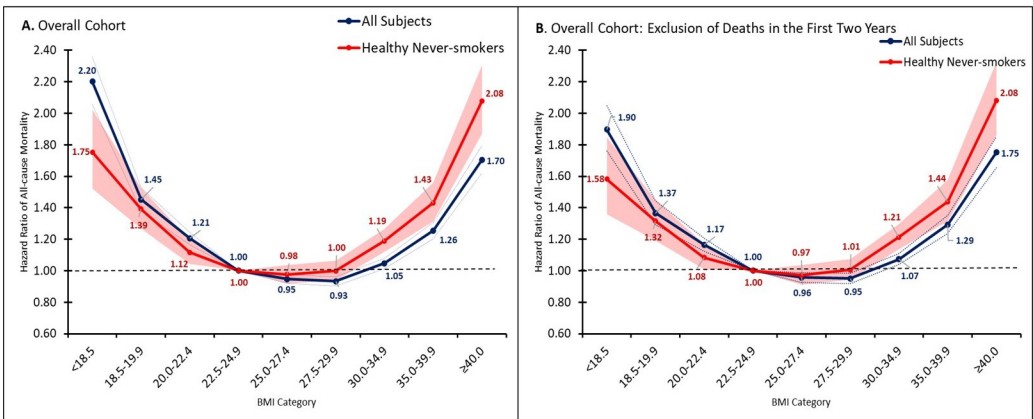

**Fig 1. Association between BMI and all-cause mortality in the overall NHIS cohort.** Fig 1 shows the hazard ratios for BMI categories, relative to a BMI of 22.5–24.9. Confidence bands represent 95% CI. (A) presents the hazard ratios for the overall cohort. The blue line represents the overall cohort. The red line depicts healthy, never-smoking individuals. Healthy defined as no self-reported history of cardiovascular disease or non-skin cancer except melanoma. (B) presents the hazard ratios for the overall cohort, excluding individuals who died within 2 years of follow-up. The blue line represents the overall sample. The red line depicts healthy, never-smoking individuals. Healthy defined as no self-reported history of cardiovascular disease or non-skin cancer or melanoma. Please note that hazard ratios for BMI groups for all subjects and healthy, never-smokers are relative to different reference groups and thus may not be comparable.

## BMI and mortality in subgroups

**Gender and age group.** The BMI-mortality patterns observed in the overall population remained largely the same in men and women, even after adjustment for covariates and restriction to healthy never-smokers (**Fig 2**). However, men and women aged ≥65 years had attenuated BMI-mortality associations compared to men <65 after restriction to healthy never-smokers (**Fig 3**). When stratifying by age group (≥65, 20–64) alone, we found that the decreased mortality seen from BMI of 25.0 to 29.9 was more pronounced in older adults and that younger adults had increased mortality risk (**S3 Fig in S2 Appendix**). Older adults also had significantly lower unadjusted risk of mortality among BMI of 30–34.9 compared to younger adults (HR [95% CI], BMI of 30–34.9; older adults: 0.85 [0.82, 0.89]; younger adults: 1.60 [1.52, 1.69]). These age-related differences persisted and remained statistically significant, after adjustment for covariates and after excluding participants who died within the first two years of follow-up (**S3 Fig in S2 Appendix**).

**Race/Ethnicity.** Among non-Hispanic White, non-Hispanic Black, and Hispanic participants, 5-year mortality was highest in those with BMI <18.5 (White: 40 [95% CI, 38–43] per 1,000 person-years, Black: 43 [37–50], Hispanic: 20 [16–25]). Five-year mortality was lowest at 11.7 (11.2–12.2) among those with BMI of 30.0–34.9 in White adults and 10.0 (8.8–11.3) at BMI of 35.0–39.9 in Black adults, and 7.1 [6.5–7.8] in Hispanic adults. The unadjusted risks of mortality were similar between non-Hispanic White, Black, and Asian adults but were significantly higher in Hispanic adults with BMIs from 25.0 to 29.9 (unadjusted HR [95% CI]. BMI of 25.0–27.4: White, 1.03 [0.99, 1.06]; Black, 1.06 [0.97, 1.15]; Asian, 1.07 [0.89, 1.28]; Hispanic, 1.14 [1.03, 1.25]; BMI of 27.5–29.9: White, 1.01 [0.98, 1.05]; Black, 0.98 [0.90, 1.06]; Asian, 0.95 [0.75, 1.20]; Hispanic, 1.15 [1.03, 1.29]). These patterns persisted, albeit insignificantly, after adjustment (**Fig 4,** P-interaction = 0.21).

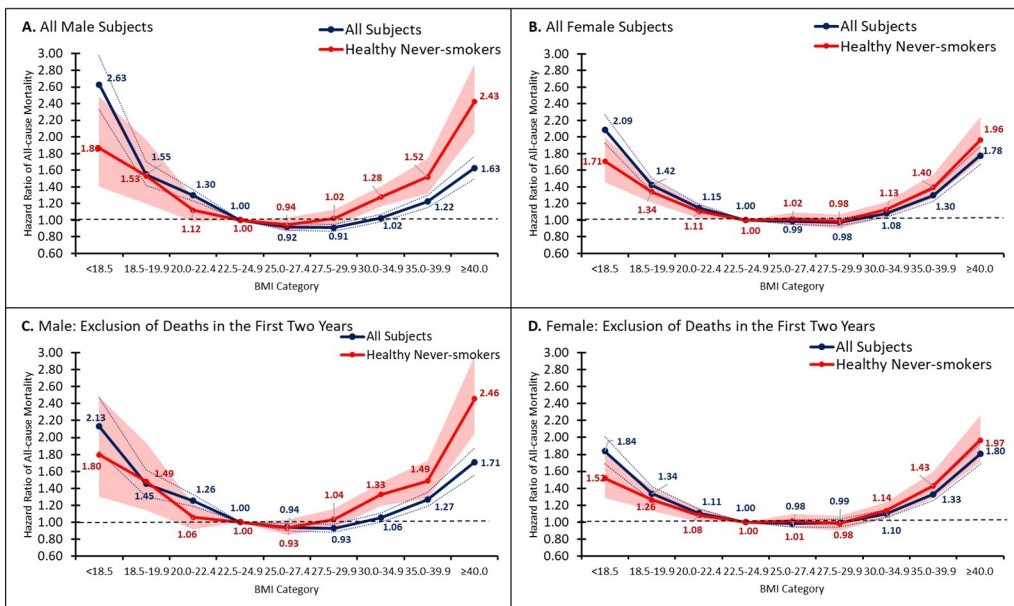

**Fig 2. Association between BMI and all-cause mortality by gender.** Fig 2 shows the hazard ratios for BMI categories, relative to a BMI of 22.5–24.9, by gender. Confidence bands represent 95% CI. The blue line represents all subjects within subgroup. The red line depicts healthy, never-smoking subjects within subgroup. Healthy defined as no self-reported history of cardiovascular disease or non-skin cancer or melanoma. (A) presents the hazard ratios among males overall. (B) presents the hazard ratios for females overall. (C) presents the hazard ratios among males, excluding individuals who died within 2 years of follow-up. (D) presents the hazard ratios among females, excluding individuals who died within 2 years of follow-up. Please note that hazard ratios for BMI groups for all subjects and healthy, never-smokers are relative to different reference groups and thus may not be comparable.

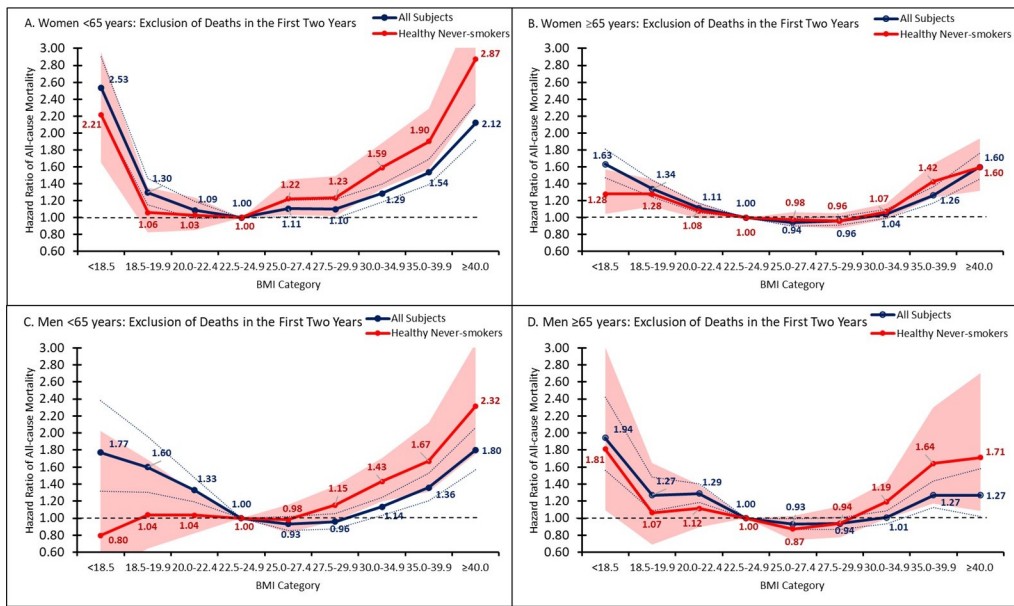

**Fig 3. Association between BMI and All-cause mortality by gender and age group.** Fig 3 shows the hazard ratios for BMI categories, relative to a BMI of 22.5–24.9, by gender and age group. All figures exclude first two years of follow-up. Confidence bands represent 95% CI. The blue line represents all individuals in subgroup. The red line depicts healthy, never-smoking individuals within subgroup. Healthy defined as no self-reported history of cardiovascular disease or non-skin cancer or melanoma. (A) presents the hazard ratios among females<65 years overall. (B) presents the hazard ratios for females greater than or equal to 65. (C) presents the hazard ratios among males<65. (D) presents the hazard ratios among males greater than or equal to 65. Please note that hazard ratios for BMI groups for all subjects and healthy, never-smokers are relative to different reference groups and thus may not be comparable.

## Sensitivity analyses in NHANES population

The NHANES sample included 44,308 adults, with a mean age of 47 years, 51% female, and 69% non-Hispanic White (**S3 Table in S2 Appendix**). Mean BMI was approximately 1 kg/m$^2$ higher in the NHANES cohort (using measured height and weight) compared to the NHIS cohort (using self-reported height and weight; **S4 Table in S2 Appendix**). In the overall NHANES population, the risk of all-cause mortality was similar across BMI groups from 22.5 to 29.9 (**S5 Table in S2 Appendix**). Unintentional weight loss significantly increased risk of mortality across all BMI categories (**S7 Table in S2 Appendix**). Adjusted HR [95% CI], BMI 25.0–27.4: 1.59 [1.25, 2.03], BMI of 27.5–29.9: 1.57 [1.19, 2.08], BMI of 30.0–34.9: 1.80 [1.42, 2.28]). Overweight and class I obese BMI participants with intentional weight loss or no weight change had similar or lower risk of mortality. The results were not sensitive to definition of 'healthy' population (**S4 Fig in S2 Appendix, S8 Table in S2 Appendix**), or inclusion of maximum lifetime BMI rather than baseline BMI in the model to account for weight history (**S9 Table in S2 Appendix**). Waist circumference (WC) modified risk of mortality, albeit insignificantly: those with overweight BMI (25.0–27.4 and 27.5–29.9) and elevated WC, but not normal WC, had higher risk of mortality, compared to participants with BMI of 22.5–24.9 with normal WC (**S6 Table in S2 Appendix**).

## Discussion

Among 554,332 U.S. adults over a recent 20-year period, the risks of all-cause mortality were similar across a wide range of BMIs including conventionally overweight BMI ranges, namely 22.5 to 29.9 kg/m$^2$. These findings persisted after adjustment for confounders, restriction to healthy never-smokers, and exclusion of participants who died within the first two years of

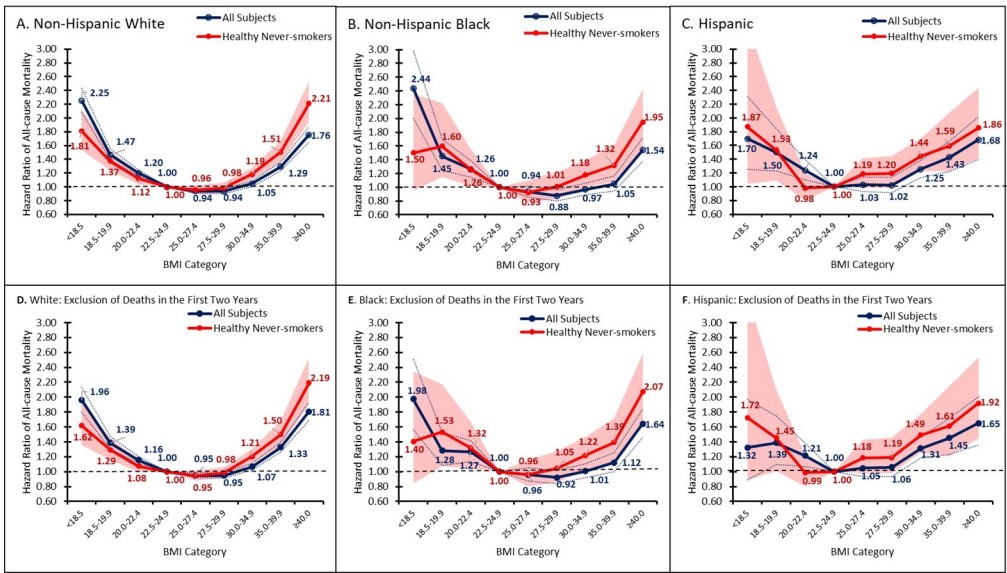

**Fig 4. Association between BMI and all-cause mortality by gender and age group.** Fig 4 shows the hazard ratios for BMI categories, relative to a BMI of 22.5–24.9, by race/ethnicity. Confidence bands represent 95% CI. The blue line represents all individuals. The red line depicts healthy, never-smoking individuals. Healthy defined as no self-reported history of cardiovascular disease or non-skin cancer or melanoma. (A) presents the hazard ratios among non-Hispanic Whites overall. (B) presents the hazard ratios for non-Hispanic Blacks. (C) presents the hazard ratios for Hispanics. (D) presents the hazard ratios among non-Hispanic Whites, excluding individuals who died within 2 years of follow-up. (E) presents the hazard ratios among non-Hispanic Blacks, excluding individuals who died within 2 years of follow-up. (F) presents the hazard ratios among Hispanics, excluding individuals who died within 2 years of follow-up. Please note that hazard ratios for BMI groups for all subjects and healthy, never-smokers are relative to different reference groups and thus may not be comparable.

follow-up [5]. Our findings were also replicated in the NHANES cohort with even better bias adjustment. Both BMI ≥30 and <18.5 were associated with significantly increased risk of mortality across all subgroups examined.

Our findings of similar all-cause mortality risks in a contemporary U.S. population over a wide range of BMI from 22.5–29.9 contrast with a pooled analysis of 19 cohort studies (15 U. S., 1976–2002) from the National Cancer Cohort Consortium [4], which found increased risk of mortality for BMI of 25.0–27.4 (HR for men: 1.06, women: 1.09) and 27.5–29.9 (men: 1.21, women: 1.19). Our results also contrast with the two largest landmark U.S. cohort studies, the NIH-AARP cohort [6,24] (data from 1995–2005) and the Cancer Prevention Study II (1982–1996) [3], both of which found 4–28% increased risk of mortality among BMI groups ≥25. Three large meta-analyses on BMI and all-cause mortality have been conducted in the past decade, utilizing studies from the 1960s through the 1990s [5,6,9]. North American results were largely driven by the aforementioned Cancer Prevention Study II and NIH-AARP cohorts. Our findings were similar to those from a meta-analysis by Flegal et al. [6] (data through 2012) and also to a representative NHANES study with data through the year 2000 [25], which both found similar or lower mortality risk for overweight BMI, although these studies used different reference BMI categories and confounding adjustments. As our statistical approach is similar to the landmark U.S. studies employing the bias adjustments recommended by the Global BMI Mortality Collaboration, the contrasting results are not likely to be explained by methodological differences.

In subgroup analyses, we found that older adults (≥65 years) demonstrated a wider range of minimally changed mortality risk than younger adults. In the NIH-AARP study, among 66–

70-year-olds, mortality risk was similar for BMI of 25.0–26.4 compared to a reference of 23.5–24.9. However, they found increased risk of mortality for BMI ≥26.5 in both men and women [24]. In contrast, a recent 2014 meta-analysis by Winter et al. showed that among adults aged ≥65 worldwide, BMI through 30.0 was associated with 4–9% decreased mortality risk compared to a reference of 23.0–23.9 [7]. Although our findings were in the same direction, they were not statistically significant (as were those of Winter et al.), likely because we performed rigorous adjustments and included a broader reference of 22.5–24.9, which may have captured a healthier reference population. Nonetheless, these cohorts, other studies in older adults [26,27], and our findings have consistently shown no significantly different mortality risk for overweight-range BMI. Among younger adults, while BMI of 25–27.4 was not associated with increased mortality, there was nearly 20% higher mortality risk for young adults with a BMI of 27.5–29.9, which is consistent with effect measures from the pooled analysis of the National Cancer Cohort Consortium.

There are several plausible reasons why participants with higher BMI (25.0–34.9) may have all-cause mortality risk similar to those with conventionally normal BMI (18.5–24.9): (1) overweight individuals may have survival advantages in various adverse circumstances, such as critical illness, major morbidities, and severe infection [28,29] that are not offset by the increased risk of chronic metabolic diseases; (2) overweight individuals without disease may be metabolically healthy and have a more favorable body composition consisting of higher lean mass [30]. Further, BMI alone may be insufficient in classifying high-risk adiposity–both waist circumference [31,32] and weight change over time can modify BMI-mortality associations, as seen in our NHANES findings; (3) lean individuals who develop diseases such as hypertension or diabetes may have more aggressive or treatment-resistant disease, whereas overweight or obese individuals who develop such conditions may be able to manage or even reverse disease with weight loss strategies [33,34]; (4) finally, the U.S. population has become much more diverse, with greater representation of ethnic minorities and older adults in our cohort with potentially different body compositions [35] compared to previous cohorts. In fact, one limitation of prior U.S. studies is a lack of racial diversity and representativeness of the changing demographic landscape. Our study included >100,000 minority adults, who exhibited lower mortality risk at overweight and obese BMIs compared to non-Hispanic White adults. This finding is consistent with data from Calle et al. [7] and with subgroup analyses from other large U.S. cohort studies [36,37].

Our study had several limitations. First, all study data in the NHIS, including BMI, was self-reported, potentially leading to misclassification [38,39]. Nevertheless, the majority of prior landmark studies have used self-reported data and have also found high correlations between findings using measured vs. self-reported BMI [7]. Further, our NHANES results, which used measured BMI, were largely similar to those from the NHIS. Second, our study had shorter follow-up than some previous studies and thus may underestimate risk of mortality in higher-BMI groups due to residual effects of occult disease. Despite the shorter follow-up, we were able to exclude early deaths without greatly compromising sample size, and more than 40% of our study participants had at least ten years of follow-up. Third, in our NHIS analysis, we analyzed BMI from a single time point. While this practice is standard across prior landmark U.S. studies, weight trajectory over time may also be relevant to the association between BMI and mortality. Our NHANES analyses using maximum lifetime BMI showed minimal change in parameter estimates. Lastly, there is a possibility of residual confounding and lack of exclusion of occult disease, as NHIS data was self-reported and likely underestimate true disease burden. However, we found that findings were sensitive to more restrictive definitions of 'healthy' individuals using medication and laboratory data in the NHANES population, suggesting that the effect of underestimated disease burden is likely to be minimal.

In conclusion, our findings suggest that BMI in the overweight range is generally not associated with increased risk of all-cause mortality. Our study suggests that BMI may not necessarily increase mortality independently of other risk factors in those with BMI of 25.0–29.9 and in older adults with BMI of 25.0–34.9. Consequently, this highlights the potential limitations of BMI in capturing true adiposity and limitations of its clinical value independent of traditional metabolic syndrome criteria. Longitudinal studies incorporating weight history, complementary measures of body composition and body fat distribution (e.g. waist circumference, waist-to-hip ratio), undermeasured consequences of weight (e.g. psychological toll of obesity [40]), and morbidity outcomes are needed to fully characterize the relationship between BMI and mortality.

## Supporting information

**S1 Appendix. Extended supplementary methods.**
(DOCX)

**S2 Appendix. S1-S4 Figs and S1-S10 Tables.**
(DOCX)

## Author Contributions

**Conceptualization:** Aayush Visaria, Soko Setoguchi.

**Data curation:** Aayush Visaria.

**Formal analysis:** Aayush Visaria.

**Investigation:** Soko Setoguchi.

**Methodology:** Aayush Visaria.

**Project administration:** Soko Setoguchi.

**Software:** Aayush Visaria.

**Supervision:** Soko Setoguchi.

**Validation:** Soko Setoguchi.

**Visualization:** Aayush Visaria.

**Writing – original draft:** Aayush Visaria, Soko Setoguchi.

**Writing – review & editing:** Aayush Visaria, Soko Setoguchi.

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
