## [Decision Letter · Decision Letter 0]

8 Nov 2022

PONE-D-22-25310

Body Mass Index and All-cause Mortality in a 21st Century U.S. Population: A National Health Interview Survey Analysis

PLOS ONE

Dear Dr. Visaria,

Thank you for submitting your manuscript to PLOS ONE. After careful consideration, we feel that it has merit but does not fully meet PLOS ONE’s publication criteria as it currently stands. Therefore, we invite you to submit a revised version of the manuscript that addresses the points raised during the review process.

We look forward to receiving your revised manuscript.

Kind regards,

Samantha Frances Ehrlich

Academic Editor

PLOS ONE

Journal Requirements:

3. Please include your tables as part of your main manuscript and remove the individual files. Please note that supplementary tables (should remain/ be uploaded) as separate "supporting information" files

Additional Editor Comments:

Thank you for this well written submission. I suggest that the Abstract should mention that the analyses were cox proportional hazards models, and agree with the Reviewer's comment that more info on the selected covariates would be appropriate. I look forward to your revision and response to the Reviewers' comments.

Reviewers' comments:

Reviewer's Responses to Questions

**Comments to the Author**

1. Is the manuscript technically sound, and do the data support the conclusions?

Reviewer #1: Yes

Reviewer #2: Yes

2. Has the statistical analysis been performed appropriately and rigorously? 

Reviewer #1: Yes

Reviewer #2: Yes

3. Have the authors made all data underlying the findings in their manuscript fully available?

Reviewer #1: Yes

Reviewer #2: Yes

4. Is the manuscript presented in an intelligible fashion and written in standard English?

Reviewer #1: Yes

Reviewer #2: Yes

5. Review Comments to the Author

Reviewer #1: Thank you for the opportunity to review “Body Mass Index and All-cause Mortality in a 21st Century U.S. Population: A National Health Interview Survey Analysis”. I commend the authors on their study in using representative data to elucidate the associations between BMI and mortality. This study provides a much-needed update to the greater scientific communities understanding of BMI and mortality. Below are my comments, questions, and suggestions.

Overall

The proposed novelty of the study is the use of a more contemporary sample of the U.S. population compared to what has been previously seen in the literature. Authors state that “nearly all U.S. studies to date used data from the 1960s through the 1990s and have included predominantly non-Hispanic White men and women.” I think that this is generally true. However, there are some studies utilizing more contemporary national surveillance data sets that also examine hazard ratios specific to race-ethnic groups (see references below). I suggest reframing the introduction to account for these three studies while also simultaneously highlighting how you study accounts for limitations of them (e.g. Linked mortality data through 2019, subgroup analyses by gender and race-ethnicity, use of a more liberal categorization of BMI to examine a more precise dose-response relationship) In other words, I believe your study does a superb job of filling in some large gaps in the literature but does not highlight its strengths as much as it should.

Nguyen et al. Characterising the relationships between physiological indicators and all-cause mortality (NHANES): a population-based cohort study. Lancet Healthy Longev 2021; 2: e651–62. Study uses data from the 1999-2014 NHANES

Zheng et al. The Body Mass Index-Mortality Link across the Life Course: Two Selection Biases and Their Effects. Uses 1999-2010 NHANES

Howell et al. Maximum Lifetime Body Mass Index and Mortality in Mexican American Adults: the National Health and Nutrition Examination Survey III (1988–1994) and NHANES 1999–2010. Uses 199-2010 NHANES data

Suggest changing subjects to particpants

Abstract

No suggestions

Introduction

See “overall” comments. The addition of the three suggested studies to your references highlights that, yes there are some contemporary population-based studies that attempt to answer this question but still have limitations that your study circumvents. A couple of sentences discussing theses studies and their limitations would strengthen your rationale.

Methods

Were the STROBE guidelines followed for this study?

Suggest providing the references that detail the linking methodology for the NDI to the NHIS.

There are some studies that suggest the use of two years for mortality during follow-up exclusion may not be enough to control for residual confounding. Did you perform any sensitivity analysis with a larger follow-up time?

5 years exclusion criteria: Di Angelantonio E, Bhupathiraju SN, Wormser D, Gao P, Kaptoge S, De Gonzalez AB, Cairns BJ, Huxley R, Jackson CL, Joshy G, Lewington S. Body-mass index and all-cause mortality: individual-participant-data meta-analysis of 239 prospective studies in four continents. The Lancet. 2016 Aug 20;388(10046):776-86.

I’m wondering if having the NHANES analysis as a secondary, sensitivity analysis diminishes its potential impact. Why not include it in your abstract and primary methods? Other studies have done this. The results from your NHANES analysis are quite impactful. It is possible that it is a space issue. Nonetheless, the NHANES analysis provides good results.

I understand the rationale for not adjusting for comorbidities in the main analysis but why not conduct subgroup analysis by disease status (e.g. CVD, cancer, diabetes, hypertension) similar to Calle et al.? It seems you might have done this in your results for diabetes and hypertension (See first comment under results heading) but there is nothing written about it. It might be prudent to see if the dose-response seen in the overall sample holds when you restrict analysis to those with disease.

Calle EE, Thun MJ, Petrelli JM, Rodriguez C, Heath Jr CW. Body-mass index and mortality in a prospective cohort of US adults. New England Journal of Medicine. 1999 Oct 7;341(15):1097-105

Results

Page six, third paragraph seems to be missing results. “Upon additionally controlling for comorbidities including diabetes and hypertension,” there are no results listed after this.

Discussion

No suggestions

Reviewer #2: This manuscript examined the risk of all-cause mortality associated with body mass index using the 1999-2018 NHIS datasets linked to the NDI. BMI, classified into 9 categories, was found to only have increased risk at or above 30 BMI.

Overall Comments. I found this study to be nicely written and of sound methodology. The results and discussion, while brief, were well organized and most appropriate. My only comments, mentioned below, focus on adding a bit more detail.

Comments

** Page 5: Covariates. I understand the need for the supplement, but I recommend that a bit more detail be added on the covariates. Just mentioning them would be fine knowing that readers can seek out the supplement to learn about the details of specific categories.

88 Page 5: Statistical Procedures. Please provide more detail on the analyses and procedures that were used within SAS.

6. PLOS authors have the option to publish the peer review history of their article (what does this mean?). If published, this will include your full peer review and any attached files.

Reviewer #1: No

Reviewer #2: No

---

## [Author Response · Author response to Decision Letter 0]

2 Feb 2023

Please see response to reviewer file. Pasted below:

Responses to Reviewer #1

1. The proposed novelty of the study is the use of a more contemporary sample of the U.S. population compared to what has been previously seen in the literature. Authors state that “nearly all U.S. studies to date used data from the 1960s through the 1990s and have included predominantly non-Hispanic White men and women.” I think that this is generally true. However, there are some studies utilizing more contemporary national surveillance data sets that also examine hazard ratios specific to race-ethnic groups (see references below). I suggest reframing the introduction to account for these three studies while also simultaneously highlighting how you study accounts for limitations of them (e.g. Linked mortality data through 2019, subgroup analyses by gender and race-ethnicity, use of a more liberal categorization of BMI to examine a more precise dose-response relationship) In other words, I believe your study does a superb job of filling in some large gaps in the literature but does not highlight its strengths as much as it should.

Thank you for your thorough review of our manuscript. It was much appreciated by our research team. We have now reframed the introduction to better address newer studies that have utilized data from the 2000-2010s, including the ones you shared. As you mentioned, some strengths of our study beyond the more recent study period include 1) subgroup analyses with sufficient power given our large sample size. NHANES data, while rich in clinical information, is far smaller than the NHIS data used in our primary analysis, and 2) comprehensive adjustment for bias and confirmation of findings in both NHIS and NHANES datasets.

Revision: 

“Of the few studies using more contemporary populations15-87, they were restricted by small sample sizes limiting analysis in racial and gender subgroups, had inadequate adjustment for methodologic bias including reverse causality, collider bias, effect modification, and the healthy person effect, and/or only rudimentarily characterized the dose-response relationship between BMI and mortality.” (Introduction, Paragraph 2, Page 3)

2. Suggest changing subjects to particpants

Thank you for this suggestion. We have now changed the term ‘subjects’ to ‘participants’ throughout the manuscript text.

3. Introduction: See “overall” comments. The addition of the three suggested studies to your references highlights that, yes there are some contemporary population-based studies that attempt to answer this question but still have limitations that your study circumvents. A couple of sentences discussing theses studies and their limitations would strengthen your rationale.

Thank you again for the suggestion. As noted above, we have changed the introduction to reflect the three studies suggested as well as one other study we found upon literature review.

4. Were the STROBE guidelines followed for this study?

Yes, we made sure to use the STROBE guidelines checklist while preparing the manuscript.

5. Suggest providing the references that detail the linking methodology for the NDI to the NHIS.

Thank you for this suggestion. We have now included the citation for the analytic file detailing the linking methodology for NDI to any National Center for Health Statistics (NCHS) survey.

Citation: National Center for Health Statistics. The Linkage of National Center for Health Statistics Survey Data to the National Death Index — 2019 Linked Mortality File (LMF): Linkage Methodology and Analytic Considerations, June 2022. Hyattsville, Maryland. Available at the following address: https://www.cdc.gov/nchs/data-linkage/mortality-methods.htm.

6. There are some studies that suggest the use of two years for mortality during follow-up exclusion may not be enough to control for residual confounding. Did you perform any sensitivity analysis with a larger follow-up time?

Thanks for this insightful comment. We did conduct several sensitivity analyses to account for reverse causality and residual confounding, including increasing the follow-up exclusion to 5 years as was done in the meta-analysis referenced. Exclusion of early deaths is a technique to try and account for subclinical/undiagnosed disease that may lead to weight loss and thus disproportionately contribute to lower BMI groups. However, we did not find any significant change in risk of mortality across any of the BMI categories. Several studies have noted similar findings – that increasing the exclusion follow-up time period may not significantly impact risks; however, it can lead to significant selection bias because a sizeable portion of the population can be excluded. We thus opted not to exclude 5 years of follow-up. To clarify this further, we explained this in more detail in the discussion and also included a supplementary table S10 that excludes the first 5 years of deaths.

Revisions:

“To examine the potential impact of reverse causality due to subclinical disease, we conducted analyses limited to participants who did not die within the first two years of follow-up. Although some prior meta-analyses have excluded the first five years of follow-up, there has been concern that such exclusions would lead to significant selection bias as a sizeable proportion of the study population would be excluded, without significant benefit in reducing reverse causality. Nevertheless, we conducted sensitivity analyses excluding the first five years of follow-up.” (Methods, Statistical Analysis, Paragraph 3, pg. 5)

See Supplementary Table S10.

7. I’m wondering if having the NHANES analysis as a secondary, sensitivity analysis diminishes its potential impact. Why not include it in your abstract and primary methods? Other studies have done this. The results from your NHANES analysis are quite impactful. It is possible that it is a space issue. Nonetheless, the NHANES analysis provides good results.

Thank you for this comment. We agree with your assessment that the NHANES results are quite impactful and provide further insight complementary to NHIS findings. However, due to word count/table limitations, we could not include it in the primary analysis and is instead in the supplement. We felt the NHIS, given its large sample size, provided much more robust estimates and allowed for subgroup analyses so it was used as the primary analysis.

8. I understand the rationale for not adjusting for comorbidities in the main analysis but why not conduct subgroup analysis by disease status (e.g. CVD, cancer, diabetes, hypertension) similar to Calle et al.? It seems you might have done this in your results for diabetes and hypertension (See first comment under results heading) but there is nothing written about it. It might be prudent to see if the dose-response seen in the overall sample holds when you restrict analysis to those with disease.

Thank you for this comment. We in fact did do subgroup analyses by disease status as determined by Calle et al., Berrington de Gonzalez et al., and our own more rigorous definition of disease. We acknowledge that this was likely not clear in the manuscript, so we have clarified this now in the revised manuscript. All figures in the main analysis have a subgroup of healthy, never-smokers who do not have CVD or cancer. In the supplementary file, Figure S2 shows the BMI-mortality associations by disease-free status. The red line depicts never-smoking individuals without minor morbidity, defined as self-reported history of cardiovascular disease, non-skin cancer except melanoma, COPD, current asthma, liver disease, kidney disease, diabetes, or functional limitations. The blue line depicts never-smoking participants without major morbidity, defined via Berrington de Gonzalez et al. as self-reported history of cardiovascular disease and non-skin cancer except melanoma. We did not show individuals with disease because of the aforementioned biases of reverse causality and confounding by pre-existing disease. Although we do not show the data, we did conduct the analysis which showed the BMI-mortality curve shifts downwards, suggesting more inverse associations with mortality at overweight BMI ranges. However, this is more likely to be due to confounding than a true inverse association.

Revision:

“We further conducted analyses among subgroups of disease-free individuals, utilizing different definitions of disease: minor morbidity, defined as self-reported history of cardiovascular disease, non-skin cancer except melanoma, COPD, current asthma, liver disease, kidney disease, diabetes, or functional limitations; and major morbidity, defined via Berrington de Gonzalez et al.4 as self-reported history of cardiovascular disease and non-skin cancer except melanoma.” (Methods, Statistical Analysis, Paragraph 2, Pg. 4)

9. Results: Page six, third paragraph seems to be missing results. “Upon additionally controlling for comorbidities including diabetes and hypertension,” there are no results listed after this.

Thank you for catching this typo. We had initially intended on deleting the sentence completely. However, we have now updated the text to complete the sentence to illustrate the concept of collider bias. This was meant to be more of a sensitivity analysis as adjustment for obesity-related comorbidities such as diabetes and hypertension would likely lead to collider bias, leading to the appearance that mortality risks are lower than they really are.

Revision:

“Upon additionally controlling for comorbidities including diabetes and hypertension (potential colliders), risk of mortality marginally decreased among overweight BMI categories (BMI 20.0-22.4: 1.10 [1.03,1.19], BMI 22.5-24.9: 1.00 [Ref], BMI 25.0-27.4: 0.95 [0.89,1.00], BMI 27.5-29.9: 0.96 [0.90, 1.02]).” (Results, Paragraph 2, Page 6)

Responses to Reviewer #2

10. Page 5: Covariates. I understand the need for the supplement, but I recommend that a bit more detail be added on the covariates. Just mentioning them would be fine knowing that readers can seek out the supplement to learn about the details of specific categories.

Thank you for this suggestion. We had included the list of covariates that we adjusted for in the Cox proportional hazards regression in the ‘Statistical Analysis’ section but realized that they were not explicitly mentioned in the Covariate subsection. We have now moved the text from the Statistical Analysis section to the Covariates subsection and provided more detail.

“Cox models were adjusted for age, gender, race/ethnicity, education, marital status, physical activity, alcohol consumption, insurance coverage, region of residence, and citizenship status.” (Methods, Statistical Analysis, Paragraph 1, pg. 4)

11. 88 Page 5: Statistical Procedures. Please provide more detail on the analyses and procedures that were used within SAS.

Thank you for this comment. We have now provided further information on the specific SAS analyses used.

---

## [Editor Report · Decision Letter 1]

16 Feb 2023

PONE-D-22-25310R1Body Mass Index and All-cause Mortality in a 21st Century U.S. Population: A National Health Interview Survey AnalysisPLOS ONE

Dear Dr. Visaria,

Thank you for submitting your manuscript to PLOS ONE. After careful consideration, we feel that it has merit but does not fully meet PLOS ONE’s publication criteria as it currently stands. Therefore, we invite you to submit a revised version of the manuscript that addresses the points raised during the review process. Please address my comments to revised manuscript (below).

We look forward to receiving your revised manuscript.

Kind regards,

Samantha Frances Ehrlich

Academic Editor

PLOS ONE

Journal Requirements:

Additional Editor Comments (if provided):

Thank you for your careful consideration of the reviewers' comments.

Please add mention that the STRBE guidelines checklist was followed in preparing this manuscript.

In the Introduction, statements pertaining to racial diversity and racial subgroups (e.g., ‘more racially diverse’) should also include ‘ethnicity’ (e.g., ‘more racially and ethnically diverse’).

There is a statement in the Introduction, ‘inadequate adjustment for methodological bias including reverse causality, collider bias, effect modification, and health person effect’. Perhaps this needs to be split into several sentences in order to clarify the points being made. For one, effect modification is not a methodological bias (though an inadequate sample size to investigate effect modification would be a limitation to prior work).

For the covariates paragraph (i.e., just prior to the ‘Statistical Analysis’ section): I suggest rephrasing ‘non-skin cancer except melanoma’ for clarity (here and throughout the manuscript, I believe this adjustment includes cancers at various sites but only melanoma for skin cancer). I also suggest including clarification of the ‘healthcare utilization factor’ variable (in parenthesis, as is done for the other covariates). Please indicate the specific supplement section where the covariate categorization is provided (and throughout the manuscript, for all mentions of ‘provided in the supplement’, please state a specific supplement section).

In the Statistical Analysis section, ‘we did not adjust for comorbidities in the main analysis, as they could be on the causal pathway between BMI and mortality or potential colliders’: Without including a DAG and explanation of potential intermediate variables at play, it is difficult to follow the potential collider argument. BMI may be causally associated with a comorbidity, such as diabetes (i.e., an arrow from BMI leading to diabetes), and diabetes causally associated with mortality (i.e., an arrow from diabetes to mortality, the causal pathway argument). However, I find it hard to justify an arrow stemming from mortality to diabetes, indicating that mortality is ‘causing’ diabetes, which would be the case if diabetes were hypothesized to be a collider here. I suggest clarifying with a DAG or removing all reference to collider bias.

Also in the Statistical Analysis section: ‘To examine the potential impact of reverse causality due to subclinical disease…’, I suggest including mention/brief clarification of what is meant by the term ‘reverse causality’, as eloquently described by your reference, Banack et al. 2019:

‘in the context of obesity-mortality research, the term reverse causality is often used to refer to a situation in which disease status affects both exposure and outcome, because disease often causes weight loss and disease increases mortality risk. Despite being called reverse causality, this is actually a concept that fits the standard definition of confounding in epidemiology…. This is why, in the context of obesity-mortality studies, the phrase reverse causality is often used interchangeably with the terms “confounding by preexisting disease” or “confounding by illness-related weight loss”

The readership of PLOS One includes epidemiologists, obesity-mortality researchers, and others, and without clarification, the language may be confusing to some.

In the Results section, characteristics: ‘Between the first four-year cycle (1999-2002) and the last cycle (2015-2018), mean BMI rose from 26.7 to 28.0 kg/m2 and prevalence of BMI of ≥30 increased from 22% to 31% (p<0.001 for trend; Supplementary Figure S1). Nearly 21% had BMI of 25.0-27.4, and 14% had BMI of 27.5-29.9.’ Supplementary Table S1 displays Mortality Rates by Subgroups, and Supplementary Table S2 displays Mortality Rates by Survey Cycle Year. Neither of these tables presents prevalence estimates directly (though these can be calculated from the data included in Table 2) so I wonder if the Supplementary Figure S1 reference here was in error? Could you clarify whether, ‘Nearly 21% had BMI of 25.0-27.4, and 14% had BMI of 27.5-29.9’ encompasses prevalence estimates for all years combined?

In the Results section, Race/ethnicity: Supplementary Table S1 does not present five-year mortality rates and unadjusted risks by race/ethnicity, please add these data or indicate ‘data not show’ (though I prefer adding the data). Did the difference between Hispanic adults with BMIs from 25.0 to 29.9 vs. other groups attain statistical significance? Was this examined with a cross product?

Discussion section: As you suggest, time period effects (e.g., differences in the racial ethnic make-up of the U.S. population over time) may explain the contrasting results, therefore mention of the time period under study for all prior work would be help to include (i.e., the meta-analysis by Flegal et al. covered what time frame/period?).

Discussion section: ‘weight change over time can both confound associations with mortality’ is confusing as written, please clarify.

Discussion section: Second to last paragraph, ‘In summary, our study of a contemporary representative….’ includes recommendations and conclusions outside of the scope of the data presented here (e.g., ‘Clinicians patients may benefit from using complementary measures of adiposity…’ and ‘guidance on management of overweight and obesity’). I suggest removing this paragraph and including mention of the limitations of BMI as a measure in the previous paragraph (or elsewhere). The last paragraph does a lovely job of summarizing the main take away points from these data and analyses.

---

## [Author Response · Author response to Decision Letter 1]

28 May 2023

Responses to Reviewer and Editor Comments:

1. Please add mention that the STRBE guidelines checklist was followed in preparing this manuscript.

Thank you for this suggestion. We have now included a sentence in the Methods section confirming we have used the STROBE guidelines checklist.

2. In the Introduction, statements pertaining to racial diversity and racial subgroups (e.g., ‘more racially diverse’) should also include ‘ethnicity’ (e.g., ‘more racially and ethnically diverse’).

Thank you for pointing out this oversight. We have now included ‘ethnicity’ in the mentioned sentence.

Revision: “In contrast, the contemporary U.S. population: (1) has a substantially different BMI distribution, with mean BMI having risen by more than 2 kg/m2 in both men and women since the 1970s, and increasing skewness towards obese-range BMIs; (2) has seen >10 year increases in life expectancy both overall and among obese individuals; (3) is more racially and ethnically diverse…” (Introduction, Page 3)

3. There is a statement in the Introduction, ‘inadequate adjustment for methodological bias including reverse causality, collider bias, effect modification, and health person effect’. Perhaps this needs to be split into several sentences in order to clarify the points being made. For one, effect modification is not a methodological bias (though an inadequate sample size to investigate effect modification would be a limitation to prior work).

Thank you for this comment. We have now changed the wording of the sentence and provided more descriptions of each of the sources of bias.

Revision: “Of the few studies using more contemporary populations, they were either restricted by small sample sizes limiting analysis in racial and gender subgroups, and/or only rudimentarily characterized the dose-response relationship between BMI and mortality. Furthermore, studies inconsistently adjusted for methodologic bias including confounding by illness-related weight loss, collider bias (e.g. the concept that conditioning on obesity-related disease may distort associations between risk factors like BMI and diet and subsequently bias downstream associations), and the healthy person effect (e.g. selection bias that may be introduced when selecting participants of different BMI groups who may otherwise be healthy).” (Introduction, Page 3)

4. For the covariates paragraph (i.e., just prior to the ‘Statistical Analysis’ section): I suggest rephrasing ‘non-skin cancer except melanoma’ for clarity (here and throughout the manuscript, I believe this adjustment includes cancers at various sites but only melanoma for skin cancer). I also suggest including clarification of the ‘healthcare utilization factor’ variable (in parenthesis, as is done for the other covariates). Please indicate the specific supplement section where the covariate categorization is provided (and throughout the manuscript, for all mentions of ‘provided in the supplement’, please state a specific supplement section).

Thank you for this comment. We have clarified the phrasing of the cancer comorbidity covariate to read as “non-skin cancer or melanoma”. We have also included a description of the healthcare utilization factors included in our analysis, and specified the specific supplement section to be referenced.

Revision: “…comorbidities (self-reported history of cardiovascular disease, non-skin cancer or melanoma, COPD, current asthma, liver disease, kidney disease, diabetes, or functional limitations), and healthcare utilization factors (doctor’s visit in the past 12 months, mental health visit in the past 12 months). Covariate categorizations are provided in the supplement (Extended Supplementary Methods)” (Methods, BMI, Mortality, and Covariates, Page 4)

5. In the Statistical Analysis section, ‘we did not adjust for comorbidities in the main analysis, as they could be on the causal pathway between BMI and mortality or potential colliders’: Without including a DAG and explanation of potential intermediate variables at play, it is difficult to follow the potential collider argument. BMI may be causally associated with a comorbidity, such as diabetes (i.e., an arrow from BMI leading to diabetes), and diabetes causally associated with mortality (i.e., an arrow from diabetes to mortality, the causal pathway argument). However, I find it hard to justify an arrow stemming from mortality to diabetes, indicating that mortality is ‘causing’ diabetes, which would be the case if diabetes were hypothesized to be a collider here. I suggest clarifying with a DAG or removing all reference to collider bias.

Thank you for this thoughtful comment. We have now included a DAG in the Supplement (Supplementary Figure S1) to illustrate the ways in which intermediary conditions such as diabetes and hypertension may be colliders. Briefly, we can posit that BMI is causally associated with diabetes and diabetes is causally associated with mortality. If there is an unmeasured confounder, such as poor nutrition, that is associated with BMI, and is causally associated with diabetes and mortality, diabetes becomes a collider (BMI and poor nutrition ‘collide’ on diabetes). Adjusting for diabetes but not for diet (as it is unmeasured) may create a spurious association between BMI and diet and affect the association between BMI and mortality through this pathway. Preston and Stokes explain this phenomenon in much greater detail and more eloquently in the following paper: Preston SH, Stokes A. Obesity paradox: conditioning on disease enhances biases in estimating the mortality risks of obesity. Epidemiology (Cambridge, Mass.). 2014 May;25(3):454.

Revision: “Consistent with prior studies, we did not adjust for comorbidities in the main analysis, as they could be on the causal pathway between BMI and mortality or potential colliders for BMI and unmeasured confounders3-5 (Supplementary Figure S1).”

Figure caption: The following figure is a crude representation of our proposed DAG for the association between BMI and all-cause mortality. We can assume BMI is causally associated with diabetes and diabetes is causally associated with mortality. If there is an unmeasured confounder, such as poor nutrition, that is associated with BMI, and is causally associated with diabetes and mortality, diabetes becomes a collider (BMI and poor nutrition ‘collide’ on diabetes). Adjusting for diabetes but not for diet (as it is unmeasured) may create a spurious association between BMI and diet and affect the association between BMI and mortality through this pathway. Preston and Stokes explain this phenomenon in much greater detail and more eloquently in the following paper: Preston SH, Stokes A. Obesity paradox: conditioning on disease enhances biases in estimating the mortality risks of obesity. Epidemiology (Cambridge, Mass.). 2014 May;25(3):454.

6. Also in the Statistical Analysis section: ‘To examine the potential impact of reverse causality due to subclinical disease…’, I suggest including mention/brief clarification of what is meant by the term ‘reverse causality’, as eloquently described by your reference, Banack et al. 2019:

‘in the context of obesity-mortality research, the term reverse causality is often used to refer to a situation in which disease status affects both exposure and outcome, because disease often causes weight loss and disease increases mortality risk. Despite being called reverse causality, this is actually a concept that fits the standard definition of confounding in epidemiology…. This is why, in the context of obesity-mortality studies, the phrase reverse causality is often used interchangeably with the terms “confounding by preexisting disease” or “confounding by illness-related weight loss”

The readership of PLOS One includes epidemiologists, obesity-mortality researchers, and others, and without clarification, the language may be confusing to some.

Thank you for this clarification. We have now changed or supplemented all reference to ‘reverse causality’ with ‘confounding by illness-related weight loss’ which we feel is definitely more intuitive to understand and can be appreciated by all readers.

Revision: “To examine the potential impact of reverse causality (i.e. confounding by illness-related weight loss), we conducted analyses limited to participants who did not die within the first two years of follow-up.” (Methods, Statistical Analysis, Page 5)

Revision: “Furthermore, studies inconsistently adjusted for methodologic bias including confounding by illness-related weight loss, collider bias…” (Introduction, Page 3)

7. In the Results section, characteristics: ‘Between the first four-year cycle (1999-2002) and the last cycle (2015-2018), mean BMI rose from 26.7 to 28.0 kg/m2 and prevalence of BMI of ≥30 increased from 22% to 31% (p<0.001 for trend; Supplementary Figure S1). Nearly 21% had BMI of 25.0-27.4, and 14% had BMI of 27.5-29.9.’ Supplementary Table S1 displays Mortality Rates by Subgroups, and Supplementary Table S2 displays Mortality Rates by Survey Cycle Year. Neither of these tables presents prevalence estimates directly (though these can be calculated from the data included in Table 2) so I wonder if the Supplementary Figure S1 reference here was in error? Could you clarify whether, ‘Nearly 21% had BMI of 25.0-27.4, and 14% had BMI of 27.5-29.9’ encompasses prevalence estimates for all years combined?

Thank you for pointing this out. Supplementary Figure S1 (now Supplementary Figure S2) Panel C shows the distribution of BMI from the first four-year cycle to the last. We realize that it is not possible to calculate the mean BMI or the percent increase in BMI>=30 visually from that so we have now removed reference to Figure S1. The statement, “Nearly 21% had BMI of 25.0-27.4, and 14% had BMI of 27.5-29.9’ encompasses prevalence estimates for all years combined” refers to the prevalence for all years combined and is from Table 1.

Revision: “Between the first four-year cycle (1999-2002) and the last cycle (2015-2018), mean BMI rose from 26.7 to 28.0 kg/m2 and prevalence of BMI of ≥30 increased from 22% to 31% (p<0.001 for trend; see Supplementary Figure S2 for distribution of BMI). Nearly 21% had BMI of 25.0-27.4, and 14% had BMI of 27.5-29.9 (Table 1).” (Results, Sample Characteristics, Page 5)

In the Results section, Race/ethnicity: Supplementary Table S1 does not present five-year mortality rates and unadjusted risks by race/ethnicity, please add these data or indicate ‘data not show’ (though I prefer adding the data). Did the difference between Hispanic adults with BMIs from 25.0 to 29.9 vs. other groups attain statistical significance? Was this examined with a cross product?

Thanks for pointing out this oversight. We had included some data in the supplementary text underneath Supplementary Table S1 that we have now included in the main text: “Among non-Hispanic White, non-Hispanic Black, and Hispanic participants, 5-year mortality was highest in those with BMI <18.5 (White: 40 [95% CI, 38-43] per 1,000 person-years, Black: 43 [37-50], Hispanic: 20 [16-25]). Five-year mortality was lowest at 11.7 (11.2-12.2) among those with BMI of 30.0-34.9 in White adults and 10.0 (8.8-11.3) at BMI of 35.0-39.9 in Black adults, and 7.1 [6.5-7.8] in Hispanic adults.”

The multiplicative interaction (cross-product) between race/ethnicity and BMI on all-cause mortality did not reach significance after adjustment for covariates. However, we felt it was still important to include racial/ethnic subgroups because of the well-studied racial/ethnic differences in various components of body composition and CV risk factors. We have now more explicitly stated that in the text.

Revision: “These patterns persisted, albeit insignificantly, after adjustment (Figure 4, P-interaction = 0.21).” (Results, Race/ethnicity, Page 6)

8. Discussion section: As you suggest, time period effects (e.g., differences in the racial ethnic make-up of the U.S. population over time) may explain the contrasting results, therefore mention of the time period under study for all prior work would be help to include (i.e., the meta-analysis by Flegal et al. covered what time frame/period?).

Thank you for this comment. We have now included the time periods for all the referenced studies in the discussion section.

9. Discussion section: ‘weight change over time can both confound associations with mortality’ is confusing as written, please clarify.

Apologies for the confusing wording. We have now changed the wording slightly to have it read better.

Revision: “Further, BMI alone may be insufficient in classifying high-risk adiposity – both waist circumference31,32 and weight change over time can modify associations BMI-mortality associations, as seen in our NHANES findings” (Discussion, Page 8)

10. Discussion section: Second to last paragraph, ‘In summary, our study of a contemporary representative….’ includes recommendations and conclusions outside of the scope of the data presented here (e.g., ‘Clinicians patients may benefit from using complementary measures of adiposity…’ and ‘guidance on management of overweight and obesity’). I suggest removing this paragraph and including mention of the limitations of BMI as a measure in the previous paragraph (or elsewhere). The last paragraph does a lovely job of summarizing the main take away points from these data and analyses.

Thank you for this comment. We have adjusted the final two paragraphs of the discussion to exclude any out-of-scope implications and combined them into one paragraph.

Revision: “In conclusion, our findings suggest that BMI in the overweight range is generally not associated with increased risk of all-cause mortality. Our study suggests that BMI may not necessarily increase mortality independently of other risk factors in those with BMI of 25.0-29.9 and in older adults with BMI of 25.0-34.9. Consequently, this highlights the potential limitations of BMI in capturing true adiposity and limitations of its clinical value independent of traditional metabolic syndrome criteria. Longitudinal studies incorporating weight history, complementary measures of body composition and body fat distribution (e.g. waist circumference, waist-to-hip ratio), undermeasured consequences of weight (e.g. psychological toll of obesity), and morbidity outcomes are needed to fully characterize the relationship between BMI and mortality.” (Discussion, Page 8-9).

---

## [Editor Report · Decision Letter 2]

2 Jun 2023

Body Mass Index and All-cause Mortality in a 21st Century U.S. Population: A National Health Interview Survey Analysis

PONE-D-22-25310R2

Dear Dr. Visaria,

We’re pleased to inform you that your manuscript has been judged scientifically suitable for publication and will be formally accepted for publication once it meets all outstanding technical requirements.

Kind regards,

Samantha Frances Ehrlich

Academic Editor

PLOS ONE

Additional Editor Comments (optional):

Thank you for carefully addressing our comments.
---

## [Editor Report · Acceptance letter]

9 Jun 2023

PONE-D-22-25310R2 

Body Mass Index and All-cause Mortality in a 21st Century U.S. Population: A National Health Interview Survey Analysis 

Dear Dr. Visaria:

I'm pleased to inform you that your manuscript has been deemed suitable for publication in PLOS ONE. Congratulations! Your manuscript is now with our production department. 

Kind regards, 

on behalf of

Dr. Samantha Frances Ehrlich 

Academic Editor

PLOS ONE